# Dynamic Changes in Plant Secondary Metabolites Induced by *Botrytis cinerea* Infection

**DOI:** 10.3390/metabo13050654

**Published:** 2023-05-13

**Authors:** Zhaochen Wu, Tuqiang Gao, Zhengya Liang, Jianjun Hao, Pengfei Liu, Xili Liu

**Affiliations:** 1Department of Plant Pathology, China Agricultural University, Beijing 100193, China; zhaochenwu@cau.edu.cn (Z.W.); gaotq@cau.edu.cn (T.G.); lzyabd@cau.edu.cn (Z.L.); seedling@cau.edu.cn (X.L.); 2School of Food and Agriculture, University of Maine, Orono, ME 04469, USA; jianjun.hao1@maine.edu

**Keywords:** metabolomics, antagonistic activity, fungicide resistance

## Abstract

In response to pathogen infection, some plants increase production of secondary metabolites, which not only enhance plant defense but also induce fungicide resistance, especially multidrug resistance (MDR) in the pathogen through preadaptation. To investigate the cause of MDR in *Botrytis cinerea,* grapes ‘Victoria’ (susceptible to *B. cinerea*) and ‘Shine Muscat’ (resistant to *B. cinerea*) were inoculated into seedling leaves with *B. cinerea*, followed by extraction of metabolites from the leaves on days 3, 6, and 9 after inoculation. The extract was analyzed using gas chromatography/quadrupole time-of-flight mass (GC/QTOF) combined with solid-phase microextraction (SPME) for volatile and nonvolatile metabolomic components. Nonvolatile metabolites γ-aminobutyric acid (GABA), resveratrol, piceid, and some carbohydrates or amino acids, coupled with volatile metabolites β-ocimene, α-farnesene, caryophyllene, germacrene D, β-copaene, and alkanes, accumulated at a higher level in grape leaves infected with *B. cinerea* compared to in noninoculated leaves. Among the established metabolic pathways, seven had greater impacts, including aminoacyl-tRNA biosynthesis, galactose metabolism, valine, leucine, and isoleucine biosynthesis. Furthermore, isoquinoline alkaloid biosynthesis; phenylpropanoid biosynthesis; monobactam biosynthesis; tropane, piperidine, and pyridine alkaloid biosynthesis; phenylalanine metabolism; and glucosinolate biosynthesis were related to antifungal activities. Based on liquid chromatography/quadrupole time-of-flight mass (LC/QTOF) detection and bioassay, *B. cinerea* infection induced production of plant secondary metabolites (PSMs) including eugenol, flavanone, reserpine, resveratrol, and salicylic acid, which all have inhibitory activity against *B. cinerea.* These compounds also promoted overexpression of ATP-binding cassette (ABC) transporter genes, which are involved in induction of MDR in *B. cinerea*.

## 1. Introduction

Plant pathogens interact with their hosts in several ways in the infection course. One consequence of this interaction is that a pathogen will elicit production of plant secondary metabolites (PSMs), such as terpenoids, phenylpropanoids, fatty acids, and alkaloids. These compounds protect plants from biotic and abiotic stresses [1].

Plant secondary metabolites may be either preformed and constitutively produced in healthy plants or synthesized in response to pathogen infection [2]. *Arabidopsis thaliana* [3,4], *Ephedra breana*, *Fabiana imbricata*, *Nolana sedifolia* [5], *Origanum syriacum* var. *bevanii* L, *Lavandula stoechas* L. subsp. *stoechas*, *Rosmarinus officinalis* L. [6], *Eugenia caryophyllata*, and *Cinnamomum zeylanicum* [7] have been reported to produce high amounts of PSMs triggered by *B. cinerea* infection. Similarly, grapes (*Vitis* spp.) possess or produce bioactive compounds [8]. It has been frequently reported that *B. cinerea* infection is associated with type and intensity of photosynthesis [9,10]. Furthermore, C3 plants are less resistant than β-carboxylating CAM plants. Low concentrations of reactive oxygen species (ROSs) induced by biotic and abiotic stresses can act as signaling molecules to regulate development processes and defense responses in host plants [9,11]. These differences are related to production of PSMs [11]. PSMs are not only involved in signal regulation of pathogen–plant interaction but also may have direct antimicrobial activity against *B. cinerea* [3,4,5,6,7,8,11].

Unfortunately, PSMs have bidirectional effects on invading pathogens; some PSMs have antimicrobial activities, while some PSMs induce pathogen and insect resistance to antibiotics and pesticides [12,13,14,15]. Continuous exposure to PSMs may result in a pest becoming adapted to the phytochemicals (preadaptation) in response to drugs and other adverse conditions. Plant secondary metabolites can elicit insects to develop detoxification through preadaptation [12]. Detoxification enzyme systems of insects include but are not limited to P450 mono-oxygen transferases [12], carboxylesterases [13], glutathione transferases [16], ATP-binding cassette (ABC) transporter proteins [17], and uridine diphosphate (UDP)-glucuronosyltransferases [18].

*Botrytis cinerea* is a widely distributed plant pathogen that attacks many economically important plants, such as grapes, resulting in significant losses in production [19]. Although *B. cinerea* has mechanisms to interfere with metabolic pathways of plants by detoxifying phytoalexins in grapevines [20], the documentation on whether PSMs can induce pathogen preadaptation is limited. We have recently confirmed that PSMs can induce preadaptation-based multidrug resistance (MDR) in *B. cinerea* (unpublished). The objectives of this study were to identify volatile and nonvolatile indicator compounds in grapes during infection and discover key PSMs that have antimicrobial effects against *B. cinerea* and induce MDR in the fungus.

## 2. Materials and Methods

### 2.1. Chemicals

Tween 20 was purchased from Beijing Solarbio Science & Technology Co., Ltd. (Beijing, China). Methanol and acetonitrile of HPLC grade were purchased from Sinopharm Chemical Reagent Co., Ltd. (Shanghai, China). Pyridine, methoxyamine hydrochloride, N, and O-bis (trimethylsilyl) trifluoroacetamide (BSTFA) (containing 1% trimethylchlorosilane, TMS), all analytical-grade, were purchased from Sigma-Aldrich (St. Louis, MO, USA). Ultrapure water was obtained from a Milli-Q system (Millipore, Billerica, MA, USA). Standards with concentrations of more than 99% were purchased from Sigma-Aldrich. Technical-grade resveratrol (a.i. 99%), reserpine (98%), and eugenol (99%) were purchased from Shanghai Maclean Biochemical Technology Co., Ltd. (Shanghai, China). Salicylic acid was provided by Beijing Soleibao Science & Technology Co., Ltd. (Beijing, China). Flavanone (98%) was obtained from Tokyo Chemical Industry Co. (Tokyo, Japan). All chemicals were dissolved in dimethyl sulfoxide (DMSO) to prepare a stock solution (1 × 10^5^ µg/mL) and kept in darkness at −20 °C.

### 2.2. Plants and Pathogens

Grape leaves were selected as the experimental material since they are susceptible tissues to *B. cinerea* infection. Grapes ‘Victoria’ (sensitive to *B. cinerea*) and ‘Shine Muscat’ (resistant to *B. cinerea*) were grown in a greenhouse from February to April 2022. *Botrytis cinerea* multidrug-resistant strain 242 was collected from Yinlong Soil Farm in Shanghai in April 2016. *Botrytis cinerea* standard strain B05.10 was obtained from Germany (Institut für Botanik, Westfälische Wilhelms-Universität, Münster). The two strains were cultivated on potato dextrose agar (PDA) in the dark at 18 °C. To prepare spore suspension as an inoculum, the fungi were cultured in the dark at 18 °C on carrot agar (CA) dishes [21] for 3 to 5 d and placed under a blacklight lamp for 5 d. Conidia were washed off the culture with sterile water and filtered through two-layer lens-wiping papers to remove mycelia. The conidial concentration was adjusted to 5 × 10^6^/mL with a hemacytometer. The suspension with 1.5% glucose and 0.5% tween 20 (Beijing Soleibao Technology Co., Ltd., Beijing, China) was also added.

### 2.3. Nonvolatile Metabolites of Grape Leaves

Grape seedlings were either inoculated, by spraying *B. cinerea* B05.10 and 242 at 10^6^ conidia/mL, or noninoculated. The treatments included ‘Victoria’ without inoculation, inoculated with B05.10, and inoculated with 242 and ‘Shine Muscat’ without inoculation, inoculated with B05.10, and inoculated with 242. Disease severity and incidence were evaluated at the end of the trial, followed by collection of all plant samples into a Ziplock bag that was stored at −80 °C for later use.

The sampled seedlings were processed for metabolomic analysis following the previously reported procedure [22]. Briefly, the sample was prechilled with liquid nitrogen and ground in a ball mill (MM400; Verder Shanghai Instruments and Equipment Co., Ltd., Shanghai, China) at 30 Hz for 1 min. Each treatment had four replicates (plants). Each sample (200 ± 2 mg) was dissolved in 1.8 mL of extraction solvent (methanol/water, *v*/*v* = 8/2) with 10 µg/mL of ribitol as an internal standard and was distributed into 4 tubes as 4 technical replicates. Each sample was treated with 100 Hz ultrasound for 20 min, followed by centrifugation at 13,800× *g* for 15 min; 0.4 mL of supernatant was collected, desiccated at 45 °C in a vacuum concentrator, and stored at −20 °C until use. Methoxyaminen hydrochloride solution (20 mg/mL) in 100 µL was added to the sample, which was incubated at 30 °C for 2 h. The sample was added to 100 µL containing BSTFA (1% TMS) and incubated on a dry-bath block at 37 °C for 6 h for derivatization. After centrifugation, 120 µL of liquid supernatant was transferred into a sample vial sealed with a rubber cap, and metabolome analysis was performed within 48 h.

Metabolites of grape leaves were separated and detected using an HP-5MS capillary column (30 m × 0.25 mm × 0.25 µm) coupled with the 7890A-5975C GC-MS system (Agilent, CA, USA). Helium was used as a carrier gas, with a 1.1 mL/min flow rate. Each injection volume was 1 µL. A GC oven was heated to 60 °C for 1 min, then raised to 325 °C at 10 °C/min for 2 min. The auxiliary heater was set at 290 °C. The ion source (EI) temperature was set to 250 °C. Electron impact ionization (70 eV) was set in a full scan mode (m/z 50 to 600) to 0.2 s/scan.

To analyze metabolites using liquid chromatography–mass spectrometry (LC-MS), the sample was prechilled with liquid nitrogen and ground in a ball mill (MM400; Verder Shanghai Instruments and Equipment Co., Ltd., Shanghai, China) at 30 Hz for 1 min. For each treatment, six samples were collected as replicates; 500 mg of sample was dissolved in 3 mL of extraction solvent (methanol/acetonitrile, *v*/*v* = 3/1) and divided into 6 technical replicates (6 tubes). Each sample was vortexed at 2500 rpm for 2 min, followed by centrifugation at 13,800× *g* for 15 min at 4 °C, and then filtered through a 0.22 μm filter membrane.

LC-MS chromatographic analyses were performed on a Thermo Vanquish UHPLC system coupled with an ACQUITY UPLC HSS T3 (2.1 mm× 150 mm, 1.7 µm) column and Thermo QE HF-X ESIs. For positive ion analysis, the eluent was a mixture of methanol and H_2_O at a flow rate of 0.3 mL/min. The mobile phase consisted of 0.1% formic acid/H_2_O (solvent A), 0.1% formic acid/methanol (solvent B), 0.1 acetic acid/H_2_O (solvent C), and 0.1% acetic acid/methanol (solvent D). Gradient steps for positive ion analysis were applied as follows (using solvents A and B): 0 to 0.5 min, B: 2%; 0.5 to 6 min, B: 2% to 50%; 6 to 10 min, B: 50% to 98%; 10 to 14 min, B: 98%; 14 to 16 min, B: 98% to 2%; 16 to 21 min, B: 2%. For negative ion analysis, solvent C was used to replace A and D to replace B. The MS system was operated using both positive and negative ESIs in multiple reaction monitoring (MRM) mode to identify the analytes of interest. The operational parameters for MS analysis were as follows: ESI source, sheath gas flow at 50 L/min and auxiliary gas at 13 L/min; spray voltage at 2.5 KV (+)/2.5 KV (−); a capillary temperature maintained at 325 °C; an auxiliary gas temperature of 300 °C; secondary collision energy (NCE) at 30 V; an isolation window at 1.5 m/z; top N = 10; and a scan range from 70 to 1050 m/z. The detected metabolites were qualified using CD software and analyzed with the Thermo Fisher database, the NIST database, and a self-built database.

### 2.4. Volatile Metabolites of Grape Leaves

Solid phase microextraction–gas chromatography–mass spectrometry (SPME-GC-MS/MS) was applied to analyze volatile metabolites. To extract volatiles, healthy grape leaves with similar sizes were selected. A 500 mL beaker was lined with cotton and tin foil, with an appropriate amount of distilled water added at the bottom to wet the cotton. The petiole of a grape leaf was inserted into the wet cotton, along with a 5 mm mycelial disk of *B. cinerea*. Volatile gases were collected at 3, 6, and 9 dpi using solid-phase microextraction (SPME) coupled with 50/30 μm DVB/CAR/PDMS fiber after aging at 250 °C for 30 min. All samples were analyzed in triplicate.

The extracted samples were examined with a GC-MS system, as previously described [23]. The program was set as splitless with 50 mL/min, an injection port of 250 °C and an ion source temperature of 230 °C. The oven program was 40 °C for 5 min and 5 °C/min to 180 °C (33 min), then 10 °C/min to 280 °C. Electron impact ionization (70 eV) was set in a full scan mode (m/z 50 to 300) to 0.3 s/scan.

### 2.5. Effects of Secondary Metabolites on Botrytis cinerea Growth

Secondary metabolites eugenol, flavanone, reserpine, resveratrol, and salicylic acid were used to examine inhibitor activity against *B. cinerea* at final concentrations of 10, 25, 50, and 100 mg/L, with three replicates, following the Song et al. method [24]. The expressions of the three major ABC genes *BcatrB*, *BcatrD,* and *BcatrK* were assessed through a quantitative polymerase chain reaction (qPCR) conducted on an ABI7500 sequence detection system (Applied Biosystems, California, USA) using the FastSYBR Mixture Kit (Beijing ComWin Biotech Co., Ltd., Beijing, China), with specific primers (Table 1) and cDNA as templates. The program of the qPCR was set as denaturation at 95 °C for 2 min, followed by 40 cycles of 95 °C for 10 s and 60 °C for 34 s. The reaction system contained 10 μL of FastSYBR × 2, 0.4/0.4 μL of a mixture of forward/reverse primer, 1 μL of template cDNA, and 8.2 μL of ddH_2_O. The template cDNA was obtained with reverse transcription from RNA of B05.10 using a PrimeScript RT Reagent Kit with gDNA Eraser (Takara, Beijing, China). The relative expression of the genes was calculated using the 2^−ΔΔCt^ method [25], and the actin gene was used as a reference to normalize the quantification of the ABC gene-expression levels. This experiment was conducted twice, and each experiment included three replicates for each treatment.

### 2.6. Statistical Analysis

Mass Hunter Qualitative Analysis B.07.00 (Agilent Technologies, Santa Clara, CA, USA) was used for data processing and deconvolution, with the parameter of 20,000 absolute peak height, and the NIST14 and Fiehn mass spectrometry databases were used as references for qualitative analysis. Agilent MassHunter Mass Profiler Professional 13.1.1 was used for principal component analysis (PCA), cluster analysis, and variance analysis. The results of the LC-MS were analyzed with Xcalibur 4.1 (Thermo Fisher Scientific, Inc., Waltham, MA, United States) and Compound Discovery. Metabo Analyst online analysis software (https://www.metaboanalyst.ca/home.xhtml, accessed on 1 August 2022) was used to conduct metabolic pathway analysis [26]. Analysis of variance (ANOVA) was performed, and metabolites were compared between *B. cinerea*-treated and non-treated groups. A significant difference was determined with fold changes >2 and *p* < 0.05.

## 3. Results

### 3.1. Nonvolatile Metabolites

A chromatogram was generated on the extract of the nonvolatile metabolites of grape leaves inoculated with *B. cinerea* or noninoculated using gas chromatography–mass spectrometry (GC-MS) (Figure 1a). In total, 287 to 312 compounds were deconvolved from the grape leaves, selected for absolute peak heights higher than 200,000, of which 206 metabolites were identified and normalized. Principal component analysis (PCA) was performed to explore the differential metabolites between six groups. A PCA score plot and PCA loading were performed to investigate the overall differences in the nonvolatile metabolites (Figure 2). All groups were distinctly separated. In both grape varieties, the noninoculated groups were separated from the inoculated groups by the largest distance. The distance between the 242- and B05.10-infected groups was small in the susceptible variety ‘Victoria’ but much larger in the resistant variety ‘Shine Muscat’, which showed a big difference in metabolites between the two varieties. Otherwise, 238 to 330 compounds were deconvolved from the samples in all groups through liquid chromatography–mass spectrometry (LC-MS), selected for absolute peak heights higher than 2000.

### 3.2. Characterization of Significantly Changed Nonvolatile Differential Metabolites

To further analyze the effects of different treatments on nonvolatile profiles, differential metabolites of treated groups relative to non-treated groups were determined by the criteria of fold changes >2 and *p* < 0.05. There were 52 differential metabolites in ‘Victoria’ under three treatments (Appendix A). Twenty-nine metabolites were upregulated and twenty-three were downregulated in B05.10-inoculated leaves compared to noninoculated leaves. Twenty-seven metabolites were upregulated and twenty-six were downregulated in 242-inoculated leaves compared to noninoculated leaves. Among these, 24 compounds were upregulated and 20 were downregulated after inoculation with both *B. cinerea* strains in the ‘Victoria’ variety.

There were 17 differential metabolites in ‘Shine Muscat’ under these treatments (Appendix A). In the B05.10-inoculated leaves, five metabolites were upregulated and twelve were downregulated. In 242-inoculated leaves, fourteen metabolites were upregulated and three were downregulated. Among these, only two compounds were upregulated in the ‘Shine Muscat’ variety inoculated with either of the two strains.

Differential metabolites, including amino acids, lipids, fatty acids, the tricarboxylic acid (TCA) cycle, and sucrose, were observed in both *B. cinerea*-inoculated and noninoculated leaves (Figure 1). 3-Pyridinol, butanoic acid, threitol, γ-aminobutyric acid (GABA), resveratrol, piceid, and some amino acids were upregulated in inoculated leaves of ‘Victoria’ grape compared to non-treated samples (Figure 2). Only threitol and 4-ketoglucose were upregulated in ‘Shine Muscat’ leaves compared to non-treated samples (Appendix A).

Meanwhile, 400 to 700 compounds were detected with LC-MS and 36 were identified by standards through liquid chromatography/quadrupole time-of-flight mass (LC/QTOF), which showed differences between the *B. cinerea*-inoculated and noninoculated grape leaves (Table 2). Among them, 14 compounds, including resveratrol (2.68 to 13.72 folds), piceid (1.52 to 3.35 folds), flavanone (1.16 to 7.28 folds), catechin (1.53 to 4.89 folds), and eugenol (1.21 to 2.11 folds), were upregulated after inoculation in both grape varieties.

### 3.3. Identification of Volatile Metabolites

Grape leaves inoculated with *B. cinerea* showed obvious disease symptoms compared to noninoculated leaves (Figure 3). About 163 to 219 volatile compounds were detected in ‘Victoria’ and ‘Shine Muscat’ filtrated for absolute peak heights higher than 2000 (Figure 1b). Among them, 133 metabolites were selected and normalized by excluding outliers and qualitative analysis via the NIST14 and Fiehn mass spectrometry databases in each group of samples. Principal component analysis (PCA) analysis revealed two major differences within the metabolic profiles of 12 treated groups (Figure 4). In ‘Victoria,’ the metabolites were separated by sampling time, regardless of inoculation. In ‘Shine Muscat,’ the metabolites were not separated by inoculation, but 3 and 6 d postinoculation (dpi) samples were clustered in one group and 9 dpi was in another group.

### 3.4. Crucially Changed Volatile Metabolites

In the ‘Victoria’ group, eleven different metabolites at 3 dpi, six different metabolites at 6 dpi, and twenty-nine different metabolites at 9 dpi were identified (Table 3). For the ‘Shine Muscat’ group, five different metabolites at 3 dpi, eleven different metabolites at 6 dpi, and eleven different metabolites at 9 dpi were found (Appendix A).

Germacrene D, butylated hydroxytoluene, and α-Farnesene were upregulated in ‘Victoria.’ β-Ocimene was upregulated in ‘Shine Muscat’. Alkanes, including β-ocimene, trans-β-ocimene, heptane, octadecane, decane, nonadecane, dodecane, eicosane, and tetradecane, were upregulated in the inoculated groups. β-ocimene was upregulated at 3, 6, and 9 dpi. 6,7-Dimethyl-1,2,3,5,8,8a-hexahydronaphthalene was upregulated at 3 dpi but downregulated at 6 dpi. On the contrary, α-Farnesene was downregulated at 3 dpi but upregulated afterward. Meanwhile, terpenes, including β-Pinene, were upregulated at 3 dpi.

### 3.5. Effect of Botrytis cinerea on Metabolic Pathways of Grapes

Metabolic pathways in grape seedlings were analyzed using the platform at Metabo Analyst. A total of 77 out of 92 nonvolatile differential compounds were categorized into 40 networks of plant metabolic pathways (Table 4). The schematic diagram indicated a global disturbance of metabolomes in grape leaves under infection of *B. cinerea*. The regulated metabolites were involved in forty-one pathways, among which seven had impacts greater than 0.1, such as aminoacyl-tRNA biosynthesis; galactose metabolism; and valine, leucine, and isoleucine biosynthesis. Among these pathways, isoquinoline alkaloid biosynthesis; phenylpropanoid biosynthesis; monobactam biosynthesis; tropane, piperidine, and pyridine alkaloid biosynthesis; phenyl-alanine metabolism; and glucosinolate biosynthesis were important in producing antifungal compounds. Moreover, there were two different networks of volatile differential metabolic pathways in the two cultivars, which were sulfur metabolism and sesquiterpenoid and triterpenoid biosynthesis in ‘Victoria’ and monoterpenoid biosynthesis and sesquiterpenoid and triterpenoid biosynthesis in ‘Shine Muscat’.

### 3.6. Bioactivity of Major Metabolites on Botrytis cinerea

Eugenol, flavanone, reserpine, resveratrol, and salicylic acid inhibited *B. cinerea* (Figure 5) and promoted overexpression of the ABC transporter genes *BcatrB*, *BcatrD,* and *BcatrK*. *BcatrB* in B05.10-inoculated grapes treated with resveratrol and eugenol was overexpressed forty-fivefold more than in non-treated plants (Figure 6).

## 4. Discussion

We found a trend that *B. cinerea* infection induced production of several PSMs to be upregulated from those in noninoculated grape leaves, although the increase was moderate. In inoculated leaves, many volatile PSMs were upregulated, such as β-ocimene, α-farnesene, caryophyllene, germacrene D, β-copaene, and alkanes. Nonvolatile PSMs showed a complex network of relationships. These included GABA, resveratrol, piceid, and some amino acids. Among them, eugenol, flavanone, reserpine, resveratrol, and salicylic acid were highly upregulated in two varieties of grapes inoculated with both isolates of *B. cinerea*. These compounds, with antifungal activities against *B. cinerea*, promoted overexpression of ABC transporter genes, which is associated with induction of MDR of *B. cinerea*. These findings are supported by previous results [31,32].

By comparing the metabolites and their pathways in grape leaves inoculated and noninoculated with *B. cinerea*, we found that they had significant differences in composition of volatile PSMs and predominant formation pathways. Some of these PSMs with antimicrobial activities, such as alkenes including heptane, octadecane, decane, nonadecane, dodecane, eicosane, tetradecane, β-ocimene, α-farnesene, caryophyllene, germacrene D, and β-copaene and the terpenoid compound β-pinene were more abundant [7,33,34,35,36,37,38,39,40]. Inoculated leaves also accumulated salicylic acid (SA) and jasmonic acid (JA). Both SA and JA are key signaling molecules involved in systemic acquired resistance (SAR) against biotrophic and necrotrophic pathogens. This is supported by previous studies [9,41]. β-ocimene triggers plant defense responses via the signaling pathways of SA, JA, and ethylene (ET) [3,42]; increases concentrations of some glucosinolates involved in plant immunity [43]; and has antifungal activity [33]. (E, E)-α-farnesene is associated with disease inhibition demonstrated in fungal pathogens, suggesting its production is involved in fungal pathogenesis [34,35,36]. All of these results indicate that PSM upregulation mechanisms follow induction of *B. cinerea* infection.

*Botrytis cinerea* infection induced or increased upregulation of nonvolatile compounds, including 3-pyridinol, butanoic acid, threitol, GABA, resveratrol, piceid, flavanone, catechin, eugenol, and some amino acids, regardless of grape variety. γ-aminobutyric acid (GABA) plays as a signal molecule, eliciting plant defense against abiotic stresses [44], as it is involved in the expressions of genes for plant signal transduction, transcriptional control, hormone biosynthesis, reactive oxygen species generation, and polyamine metabolism [31,43,44]. Therefore, GABA is an important metabolite, contributing to disease resistance. The Lys catabolite pipecolic acid (Pip) is a critical metabolic mediator of several forms of inducible resistance in Arabidopsis, which accumulates following pathogen recognition [45]. Then, N-hydroxypipecolic acid (NHP) is the actual regulator of SAR rather than Pip [46,47]. This result was consistent with the downregulated Pip in the leaves of the treated group compared to the non-treated group, probably attributable to hydroxylation of Pip to NHP. Unfortunately, the compound directly involved in SAR was not detected.

Plants have developed several systems to regulate levels of ROSs. Abiotic stresses cause accumulation of ROSs and reduce photosynthetic activity, which triggers production of hydrogen peroxide (H_2_O_2_) and compounds such as proline, glutathione, ascorbic acid, carotenoids, flavonoids, and tocopherols to alleviate oxidative damage by neutralizing ROSs [11]. Among the PSMs detected in this study, ascorbic acid was upregulated after *B. cinerea* infection regardless of grape variety. This indicates that ascorbic acid may act as a deoxidizer.

Plant secondary metabolites (PSMs) possessing antifungal activities can be a potential resource for fungicide development. For example, since resveratrol and its derivative are linked to plant immunity [32,48,49], they have been registered as an effective fungicide in China (http://www.chinapesticide.org.cn/, accessed on 31 December 2021) for controlling cucumber gray mold. In this study, resveratrol, reaveratroloside, and trans-piceid showed significant upregulation in *B. cinerea*-inoculated grape leaves. Resveratrol resulted in up to 47% inhibition of *B. cinerea*. This moderate efficacy can be increased by changing the chemical structure, which was demonstrated by in vivo methylation of hydroxyphenyl groups in phenolics [50].

Differential metabolites were more numerous in the resistant variety ‘Shine Muscat’ than in the susceptible variety ‘Victoria’. ‘Shine Muscat’ had more compounds with higher levels of antifungal activity regardless of *B. cinerea* inoculation. These compounds might suppress infection and proliferation of pathogens [51]. Interestingly, some metabolites were regulated differently when the grape was infected with wild-type *B. cinerea* or with an MDR-containing strain. While *B. cinerea* 242 can develop MDR, it may have a cost of fitness and reduced virulence toward the host plant, which was confirmed not only in this study but also in several other studies on *B. cinerea*. Therefore, acquisition of high-level resistance is associated with a decrease in fitness [52]. Meanwhile, grape plants with different resistance have different metabolomic activities, as shown in the susceptible variety ‘Victoria’ versus the resistant variety ‘Shine Muscat.’ This indicates that plant defense against pathogen infection through metabolomic regulation is dependent on its resistance level.

On one hand, PSMs, such as phytoalexins, can be induced by pathogens and increase plant defense against the pathogens [53,54]. On the other hand, the pathogens may be adaptive to and detoxify PSMs by enhancing the ABC efflux system [55,56,57,58]. In fungi, ABC transporters are involved in transportation of PSMs such as phenylpropanoids, lignans, flavonoids, and condensed tannins [59,60] and are also upregulated when exposed to some PSMs, such as eugenol and reserpine [61]. These studies support our results. We found that eugenol, flavanone, reserpine, resveratrol, and salicylic acid inhibited *B. cinerea* but promoted, in the fungus, overexpression of three major ABC transporter genes: *BcatrB*, *BcatrD,* and *BcatrK*. As such, we speculate that when exposed to antimicrobial PSMs, the pathogen would carry out the efflux of PSMs, impeding their antimicrobial activity. All tested PSMs exerted growth inhibition but could potentially elicit preadaptation in *B. cinerea*. 

## 5. Conclusions

In this study, we found that *B. cinerea* infection triggered grapes to increase production of some secondary metabolites that have antifungal bioactivities and therefore enhanced plant resistance to diseases such as *B. cinerea*. We identified indicator compounds in grapes during infection and discovered key secondary metabolites that are associated with the defensive activities of grapes in inhibiting *B. cinerea*. The downside of these PSMs is that the pathogen may be adaptive to and detoxify PSMs by enhancing the ABC efflux system, which impedes the antimicrobial activity of PSMs. These findings enhanced our understanding of plant resistance in host plants and provided a new perspective for development of MDR of *B. cinerea*. PSM adaptation with the pathogen should be considered when a fungicide is to be developed using PSMs and their derivatives.

## Figures and Tables

**Figure 1 metabolites-13-00654-f001:**
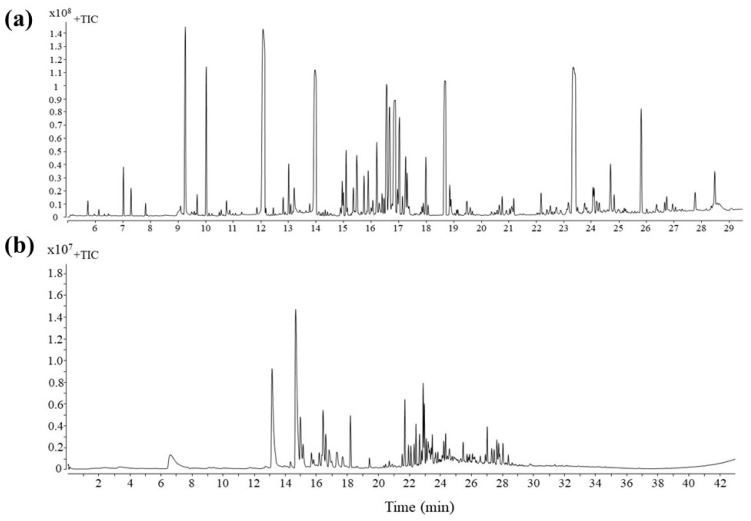
Total ion current (TIC) diagram of gas chromatography–mass spectrometry for nonvolatile metabolites (**a**) and volatile metabolites (**b**) of grape ‘Victoria’ treated with *Botrytis cinerea* strain B05.10.

**Figure 2 metabolites-13-00654-f002:**
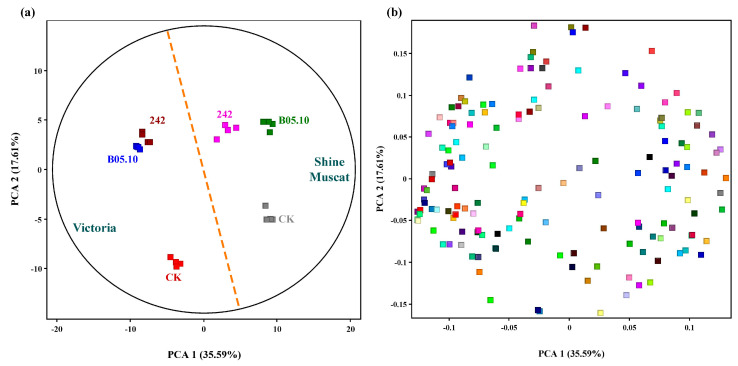
Principal component analysis (PCA) (**a**) and PCA loading score (**b**) of nonvolatile metabolomes in grape ‘Victoria’ and ‘Shine Muscat’ leaves inoculated with *Botrytis cinerea* strains B05.10 and 242. Treatments included ‘Victoria’ noninoculated (CK), ‘Victoria’ inoculated with B05.10, ‘Victoria’ inoculated with *B. cinerea* 242, ‘Shine Muscat’ noninoculated (CK), ‘Shine Muscat’ inoculated with B05.10, and ‘Shine Muscat’ inoculated with 242.

**Figure 3 metabolites-13-00654-f003:**
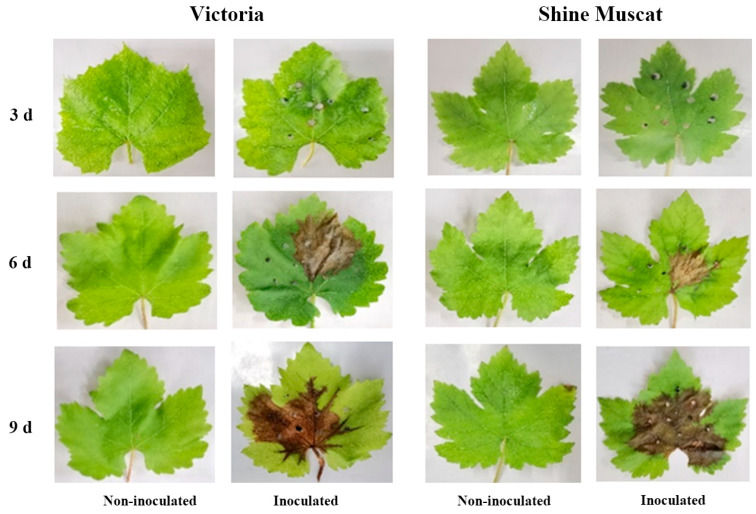
Grape leaves inoculated with *Botrytis cinerea* B05.10 at 3, 6, and 9 dpi compared with the noninoculated control.

**Figure 4 metabolites-13-00654-f004:**
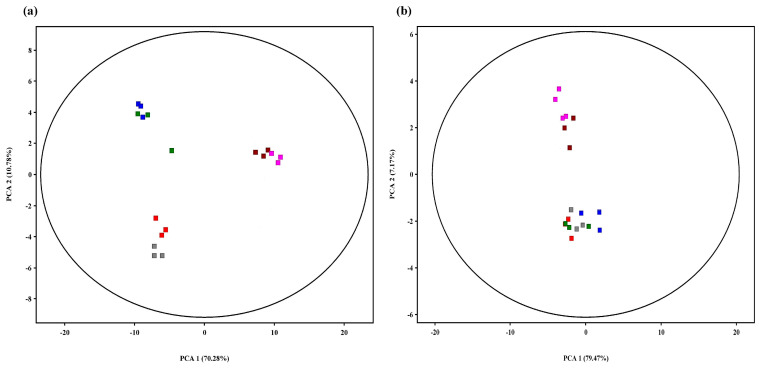
Principal component analysis (PCA) of volatile metabolomes in grapes ‘Victoria’ (**a**) and ‘Shine Muscat’ (**b**) inoculated with *Botrytis cinerea* B05.10. Treatments include noninoculation at 3 dpi (A 
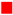
), noninoculation at 6 dpi (B 
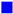
), noninoculation at 9 dpi (C 
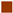
), inoculation at 3 dpi (D 
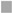
), inoculation at 6 dpi (E 
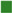
), and inoculation at 9 dpi (F 
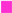
).

**Figure 5 metabolites-13-00654-f005:**
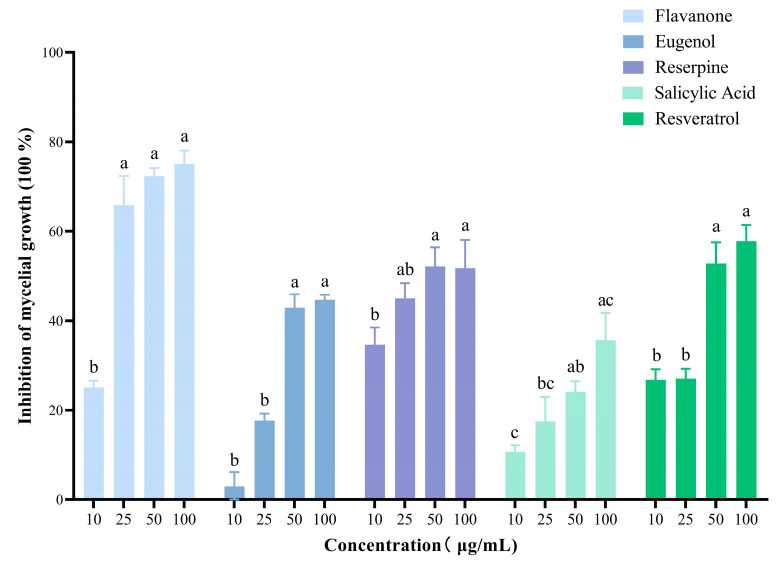
Sensitivity to flavanone, eugenol, reserpine, salicylic acid and resveratrol of *Botrytis cinerea* B05.10. Letters indicate significant difference between inhibition of mycelial growth from each compound according to two-way ANOVA.

**Figure 6 metabolites-13-00654-f006:**
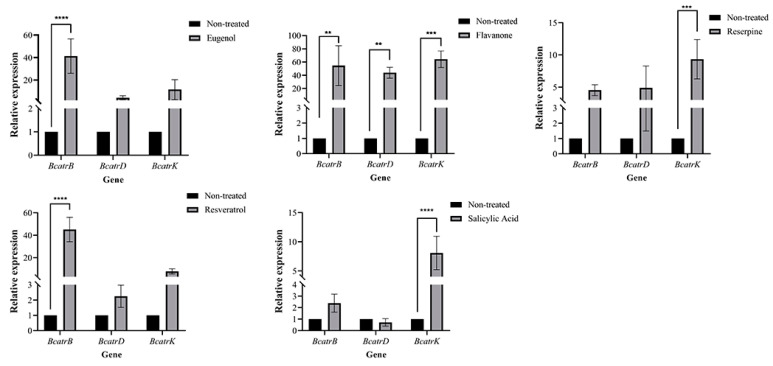
The expressions of ATP binding cassette (ABC) transporter genes *BcatrB*, *BcatrD*, and *BcatrK* in *Botrytis cinerea* strain B05.10 exposed to eugenol, flavanone, reserpine, resveratrol, and salicylic acid. **: *p* value <0.01; ***: *p* value <0.001; ****: *p* value <0.0001.

**Table 1 metabolites-13-00654-t001:** Primers used for quantitative polymerase chain reaction (qPCR).

Gene	Sequence
*BcActin*	Forward (F): TCCAAGCGTGGTATTCTTACCC
	Reverse (R): TGGTGCTACACGAAGTTCGTTG
*BcatrB*	F: TCTAACCCCGCTGAACACAT
	R: AGAGAGGGGTTGCGAATTCA
*BcatrD*	F: TCCAGGAGCCAGCAATACAA
	R: AACCCTGCGGCAAATGAATT
*BcatrK*	F: CCGCTTTGATGGAGAACGAG
	R: GTGATGTAGTCGCCACCAAC

**Table 2 metabolites-13-00654-t002:** Differential nonvolatile metabolites of grapes ‘Victoria’ and ‘Shine Muscat’ with or without *Botrytis cinerea* inoculation, detected with liquid chromatography/quadrupole time-of-flight mass (LC/QTOF).

Compound Name	Retention Time	Formula	Adduct	m/z	Mass Error	MS/MS Spectrum	FC (VB5 vs. VNI) *	FC (VB2 vs. VNI)	FC (SB5 vs. SNI)	FC (SB2 vs. SNI)	FC (SNI vs. VNI)	MS/MS Reference	Concentration (µg/mL)
Flavonoids													
Kaempferol-7-O-glucoside	4.97	C_21_H_20_O_11_	[M + H]^−^	447.0967	8.53	447, 121, 149, 315	0.75	0.70	2.82	3.03	0.29	Database	
Catechin	5.69	C_15_H_14_O_6_	[M + H]^+^	291.8177	−2.19	139, 207, 123, 147	1.88	1.87	1.53	4.89	1.25	Database	
Flavanone	6.08	C_15_H_12_O_8_	[M + H]^+^	321.0603	−1.09	153, 275, 149, 139	2.68	3.43	1.16	7.28	2.23	Standards	2.488–6.123
Quercetin-3β-D-glucoside	7.53	C_21_H_20_O_12_	[M + H]^−^	463.0891	1.81	300, 301, 463, 271	1.83	1.33	0.74	2.00	0.59	Database	
Naringenin	7.54	C_15_H_12_O_5_	[M + H]^+^	273.0681	−1.33	-	3.94	1.29	0.53	0.65	5.97	[27]	
Kaempferol-3-glucuronide	8.66	C_21_H_18_O_12_	[M + H]^−^	461.2493	1.34	285, 461, 113, 229	0.88	0.88	3.65	5.49	0.28	Database	
Afzelin	9.01	C_21_H_20_O_10_	[M + H]^−^	431.1926	1.1	269, 285, 431, 59	2.07	2.06	2.12	2.50	4.61	Database	
Amino Acids													
Asparagine	0.87	C_4_H_8_N_2_O_3_	[M + H]^−^	131.0454	−5.68	114, 113, 131, 95	0.75	0.71	0.87	0.89	0.48	Database	
Aspartic Acid	0.89	C_4_H_7_NO_4_	[M + H]^−^	132.0294	−5.86	88, 132, 115, 71	0.31	0.27	1.59	1.43	0.73	Database	
Phenylalanine	4.35	C_9_H_11_NO_2_	[M + H]^+^	166.0861	−0.83	166, 120, 103, 149	0.13	0.18	1.07	3.91	0.60	Database	
Phenolic Acids and Derivatives													
Rhusflavanone	1.00	C_30_H_22_O_10_	[M + H]^+^	543.1326	5.34	-	1.91	1.83	1.06	1.11	1.45	[28]	
Salicylic Acid	5.63	C_7_H_6_O_3_	[M + H]^−^	137.0257	−5.39	-	1.39	1.64	1.98	0.91	1.48	Standards	0.026–0.0451
Trans-Piceid	5.81	C_20_H_22_O_8_	[M + H]^+^	391.1317	−1.67	-	1.62	1.63	1.27	1.44	1.51	[29]	
Psoralidin	6.58	C_20_H_16_O_5_	[M + H]^+^	337.1067	−0.83	337, 275, 235, 245	1.00	1.11	1.43	2.29	1.83	Database	
Reaveratroloside	7.60	C_20_H_22_O_9_	[M + H]^−^	405.213	1.03	243, 405, 361, 375	3.35	2.04	1.95	1.52	1.65	Database	
Dihydroresveratrol	7.61	C_14_H_14_O_3_	[M + H]^−^	229.094	0.01	-	0.62	0.65	1.07	1.03	1.10	[30]	
Resveratrol	8.23	C_14_H_12_O_3_	[M + H]^−^	227.0712	−0.45	227, 185, 202, 164	13.72	2.68	8.33	3.63	0.56	Standards	0.005–0.045
Cis-Resveratrol	8.23	C_14_H_12_O_3_	[M + H]^+^	229.086	−1.26	-	5.31	2.96	2.06	3.98	1.26	[29]	
Astragalin	8.57	C_21_H_20_O_11_	[M + H]^−^	447.1879	0.82	447, 284, 255, 300	0.74	0.68	2.39	2.80	0.34	Database	
Isorhamnetin	8.65	C_16_H_12_O_7_	[M + H]^+^	317.0651	−1.37	317, 285, 274, 257	0.90	0.64	0.88	4.39	0.77	Database	
Eugenol	9.09	C_10_H_12_O_2_	[M + H]^−^	163.0243	−3.29	148, 119, 163, 59	1.21	1.46	1.69	2.11	1.32	Standards	0.301–0.396
Organic Acids													
Malic Acid	1.23	C_4_H_6_O_5_	[M + H]^−^	133.039	−6.27	115, 133, 71, 72	1.15	1.15	1.12	1.20	0.92	Database	
Ascorbic Acid	1.34	C_6_H_8_O_6_	[M + H]^−^	175.0623	−3.51	115, 175, 87, 71	1.08	1.04	1.35	1.66	0.64	Database	
Pipecolic Acid	1.47	C_6_H_11_NO_2_	[M + H]^+^	130.0863	0.14	130, 84, 85, 67	0.55	0.45	1.29	2.16	0.71	Database	
2-Isopropylmalic Acid	5.91	C_7_H_12_O_5_	[M + H]^−^	175.0206	−2.96	175, 115, 85, 113	0.63	0.67	2.49	1.52	0.46	Database	
Epicatechin	6.50	C_15_H_14_O_6_	[M + H]^+^	291.1752	−2.18	139, 123, 147, 165	2.28	1.64	1.65	6.34	0.98	Database	
Ellagic Acid	8.43	C_14_H_6_O_8_	[M + H]^−^	300.8998	0.11	219, 208, 200, 126	0.87	0.90	1.65	1.17	1.06	Database	
Jasmonic Acid	9.25	C_12_H_18_O_3_	[M + H]^−^	209.1182	−0.74	59, 209, 133, 173	0.63	1.27	1.52	1.45	1.26	Database	
Oleanolic Acid	12.62	C_30_H_48_O_3_	[M + H]^−^	455.3537	1.28	455, 122, 130, 148	0.91	0.64	0.80	0.72	1.42	Database	
Abietic Acid	12.64	C_20_H_30_O_2_	[M + H]^−^	301.0357	1.76	301, 151, 229, 271	0.96	1.11	0.92	1.00	1.25	Database	
Sugars													
Glucose 1-phosphate	0.92	C_6_H_13_O_9_P	[M + H]^+^	261.0369	−0.71	98, 85, 97, 127	1.15	1.29	1.11	5.27	1.05	Database	
Alcohols													
Piceatannol	7.62	C_14_H_12_O_4_	[M + H]^−^	243.0664	0.99	243, 109, 198, 215	1.04	1.21	1.38	1.00	1.06	Database	
Alkaloids													
Berberine	8.03	C_20_H_17_NO_4_	[M + H]^+^	336.2181	−1.06	-	1.74	1.39	0.44	0.43	6.28	Standards	
Reserpine	9.28	C_33_H_40_N_2_O_9_	[M + H]^+^	609.2731	−0.43	-	1.13	0.94	0.93	0.16	5.74	Standards	1.125–1.248
Others													
Stearamide	12.98	C_18_H_37_NO	[M + H]^+^	284.2943	−1.99	284, 60, 88, 102	1.15	1.06	0.86	1.11	1.17	Database	

* VNI, ‘Victoria’ noninoculated; VB5, ‘Victoria’ inoculated with *Botrytis cinerea* B05.10; VB2, ‘Victoria’ inoculated with *B. cinerea* 242; SNI, ‘Shine Muscat’ noninoculated; SB5, ‘Shine Muscat’ inoculated with B05.10; SB2, ‘Shine Muscat’ inoculated with 242; FC, fold change.

**Table 3 metabolites-13-00654-t003:** Volatile metabolites of inoculated vs. noninoculated leaves of grape ‘Victoria’.

Days After Inoculation	Compound	Time	M/Z	Log_2_FC	Regulation
**3**	Heptane	12.5	57.0703	2.2	Up
	2,2,4,4-Tetramethyloctane	13.8	57.0701	1.9	Up
	β-Ocimene	14.6	145.9681	1.4	Up
	Octadecane	17.7	57.0701	17.9	Up
	2,5-Dihydroxybenzaldehyde	18.1	266.9993	−0.6	Down
	Cyclohexane	18.3	69.0701	18.4	Up
	Decane	18.5	57.0699	17.9	Up
	Tetradecane	19.3	57.0703	0.3	Up
	6,7-Dimethyl-1,2,3,5,8,8a-hexahydronaphthalene	20.6	91.0543	19.5	Up
	Eicosane	20.6	57.0703	0.6	Up
	Tricosanol	21.0	55.0547	0.5	Up
**6**	6,7-Dimethyl-1,2,3,5,8,8a-hexahydronaphthalene	20.5	91.0540	−0.1	Down
	Ylangene	24.1	105.0692	1.7	Up
	7-epi-Silphiperfol-5-ene	24.2	175.1478	20.8	Up
	2-Piperidinone	24.5	57.7010	1.6	Up
	Germacrene D	26.9	161.1326	1.2	Up
	Butylated Hydroxytoluene	27.7	205.1583	0.9	Up
**9**	Trans-β-Ocimene	14.3	145.9686	0.9	Up
	β-Ocimene	14.6	145.9681	0.9	Up
	2,4,6-Octatriene	17.2	121.1007	1.5	Up
	1,3-Cyclohexadiene	17.6	121.1022	1.0	Up
	Sulfurous Acid	21.5	57.0697	0.2	Up
	Nonadecane	22.2	57.0697	−0.3	Down
	n-Tridecan-1-ol	22.4	69.0697	−0.1	Down
	1-Iodo-2-methylundecane	22.7	57.0697	−0.1	Down
	1-Octanol	22.8	69.0700	0.9	Up
	Hexadecane	22.9	57.0697	−0.1	Down
	β-Guaiene	23.1	119.0856	0.3	Up
	Ylangene	24.2	105.0697	22.5	Up
	7-epi-Silphiperfol-5-ene	24.3	175.1478	22.5	Up
	2-Piperidinone	24.5	57.0700	2.2	Up
	Cis-13-Eicosenoic Acid	24.7	57.0697	21.2	Up
	Caryophyllene	25.4	105.0697	22.5	Up
	γ-Elemene	25.8	121.1004	20.8	Up
	isoledene	26.0	105.0698	21.5	Up
	Cedrene	26.1	57.0700	21.0	Up
	cis-Muurola-4	26.5	161.1326	21.6	Up
	γ-Muurolene	26.8	161.1326	21.8	Up
	Germacrene D	27.0	161.1326	23.9	Up
	Octadecane	27.2	57.0700	1.1	Up
	α-Farnesene	27.6	57.0709	0.6	Up
	Butylated Hydroxytoluene	27.7	205.1583	0.6	Up
	β-copaene	27.8	159.1168	21.7	Up
	1-Hexadecanol	27.9	69.0697	−19.9	Down
	Naphthalene	28.0	159.1162	2.3	Up
	1H-3a,7-Methanoazulene	29.7	57.0599	1.1	Up

**Table 4 metabolites-13-00654-t004:** Pathways of differential metabolites of grape leaves inoculated with *Botrytis cinerea*.

Pathway Name	Match Status ^a^	P ^b^	Holm p ^c^	Impact ^d^
Galactose Metabolism	5/27	0.003	0.263	0.272
Aminoacyl-tRNA Biosynthesis	6/46	0.006	0.610	0.000
Butanoate Metabolism	3/17	0.024	1.000	0.136
Pentose Phosphate Pathway	3/19	0.032	1.000	0.116
Valine, Leucine, and Isoleucine Biosynthesis	3/22	0.047	1.000	0.000
Alanine, Aspartate, and Glutamate Metabolism	3/22	0.047	1.000	0.255
Starch and Sucrose Metabolism	2/22	0.200	1.000	0.099
Isoquinoline Alkaloid Biosynthesis	1/6	0.207	1.000	0.000
Phenylpropanoid Biosynthesis	3/46	0.250	1.000	0.100
Monobactam Biosynthesis	1/8	0.266	1.000	0.000
Tropane, Piperidine, and Pyridine Alkaloid Biosynthesis	1/8	0.266	1.000	0.000
Lysine Biosynthesis	1/9	0.293	1.000	0.000
Cyanoamino Acid Metabolism	2/29	0.300	1.000	0.000
Phenylalanine Metabolism	1/11	0.346	1.000	0.471
Glycine, Serine, and Threonine Metabolism	2/33	0.357	1.000	0.120
Arginine and Proline Metabolism	2/34	0.370	1.000	0.066
Nicotinate and Nicotinamide Metabolism	1/13	0.395	1.000	0.000
Valine, Leucine, and Isoleucine Degradation	2/37	0.411	1.000	0.000
Sulfur Metabolism	1/15	0.440	1.000	0.033
Glucosinolate Biosynthesis	3/65	0.449	1.000	0.000

^a^ Match status is the number of matching metabolites over the total number of metabolites. ^b^ P: probability of enrichment analysis. ^c^ Holm P is the probability adjusted by the Holm–Bonferroni method. ^d^ Impact is the path topology value of influence.

## Data Availability

The data presented in this study are available online within this article or the Appendix A.

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
