# Peer review of "Dynamic Changes in Plant Secondary Metabolites Induced by Botrytis cinerea Infection"

_metabolites, 2023, doi:10.3390/metabo13050654_

Round 1

Reviewer 1 Report

Authors did a lot of good work regarding change in plant metabolites due to fungal infection. In general, write up is also good. However, some modifications are necessary to improve this manuscript before its publication.

1- There should be a little change in the title. Change "Dynamic Changes of Plant" to "Dynamic Changes in Plant"

2- In Abstract, reformat the sentence on lines 12-14 to make it more understandable.

3- On Line 21, What is meant by 7?

4- Lines 30-31: No need of equipment related lengthy keywords.

5- One Line 38, delete ",diseases, and" because diseases are biotic stress.

6- Line 37, change "fat" to "fatty".

7- Although a few latest references have been used in the Introduction, however, mostly old references have been cited. I, therefore, suggested the following latest references in Line 49.

Shoaib A, Abbas S, Nisar Z, Javaid A, Javed S (2022). Zinc highly potentiates the plant defense responses against Macrophomina phaseolina in mungbean. Acta Physiologiae Plantarum 44: 22.          

8- Line 73, use "Solarbio Science & Technology Co.," instead of " solarbio science & technology co.".

9- Lines 84 and 97: Use uniformly ml or mL.

10- Line 116: Uniformly write mg·mL−1 or mg/mL.

11- Line 124: Check spacing between a digit and its unit " 325 ℃ at 10℃. Also correct similar mistakes on lines 136 (1.7µm) and 142 ( 21min).

12- In Table 2, format the formulas of compounds correctly.

13- Also add latest references of 2022 and 2023 in Discussion.

14- Format references uniformly and correctly. Especially check the names of journals.

In general, English is good. However, there is need to improve the formatting of a few sentences.

Author Response

Authors did a lot of good work regarding change in plant metabolites due to fungal infection. In general, write up is also good. However, some modifications are necessary to improve this manuscript before its publication.

  • There should be a little change in the title. Change "Dynamic Changes of Plant" to "Dynamic Changes in Plant"

Response: Corrected.

  • In Abstract, reformat the sentence on lines 12-14 to make it more understandable.

Response: The sentence was modified as

“To investigate the cause of MDR in Botrytis cinerea, grape ‘Victoria’ (susceptible to B. cinerea) and ‘Shine Muscat’ (resistant to B. cinerea) were inoculated on seedling leaves with B. cinerea,”

  • On Line 21, What is meant by 7?

Response: We meant 7 metabolic pathways.

  • Lines 30-31: No need of equipment related lengthy keywords.

Response: The keywords have been modified.

  • One Line 38, delete ", diseases, and" because diseases are biotic stress.

Response: Corrected.

  • Line 37, change "fat" to "fatty".

Response: Corrected.

7- Although a few latest references have been used in the Introduction, however, mostly old references have been cited. I, therefore, suggested the following latest references in Line 49.

Shoaib A, Abbas S, Nisar Z, Javaid A, Javed S (2022). Zinc highly potentiates the plant defense responses against Macrophomina phaseolina in mungbean. Acta Physiologiae Plantarum 44: 22.

Response: The reference has been changed and updated.

8- Line 73, use "Solarbio Science & Technology Co.," instead of " solarbio science & technology co.".

Response: Corrected.

9- Lines 84 and 97: Use uniformly ml or mL.

Response: Modified as suggested on line 84.

10- Line 116: Uniformly write mg·mL−1 or mg/mL.

Response: Corrected on line 115.

11- Line 124: Check spacing between a digit and its unit " 325 ℃ at 10℃. Also correct similar mistakes on lines 136 (1.7µm) and 142 (21min).

Response: All have been checked and revised accordingly.

12- In Table 2, format the formulas of compounds correctly.

Response: The formulas have been correctly formatted.

13- Also add latest references of 2022 and 2023 in Discussion.

Response: The references have been added.

14- Format references uniformly and correctly. Especially check the names of journals.

Response: All references were checked and confirmed.

Reviewer 2 Report

The paper “Dynamic Changes of Plant Secondary Metabolites Induced by Botrytis cinerea Infection Contribute to Plant Defense to Diseases and Pathogen Resistance to Multiple Drugs submitted by Pengfei Liu et al describes a series of experiments to understand the chemical response of two varieties of grapes (Victoria and Shine Muscat) to the infection by Botrytis cinerea a widely distributed plant pathogen that attacks many economically important plants. The author analyzed volatile and non-volatile components bye a series of instrumental methods coupled to statistical analyses. The paper is in the overall well written, except for some minor misprints. The experimental part is well executed, and the discussion of the results are consistent with previous result already published in the area of plant-pathogen interaction and the role of secondary plant products as a defense mechanism. Some minor errors should be addressed. The major observation is the lack of a -Conclusions- section that permits to the reader to integrate the results and to have a perspective for future related research project.

1.- In line 73, the initial letter of -solarbio science & technology- must be in uppercase to be in concordance with other company names included in the Material and Methods section.

2.- In line 145 the word -sheath- must be replaced for the adequate term.

3.- In line 99, Being…must be changed to Beijing?

4.- A -Conclusions- section must be incorporated at the end of the paper. This section is of major relevance in a scientific paper.

After the inclusion of a -Conclusions- section and fix of the minor misprints already indicated I consider that the paper fulfills the requirements to be published in Metabolites.

The English language is properly used. Some errors were indicated.

Author Response

The paper “Dynamic Changes of Plant Secondary Metabolites Induced by Botrytis cinerea Infection Contribute to Plant Defense to Diseases and Pathogen Resistance to Multiple Drugs” submitted by Pengfei Liu et al describes a series of experiments to understand the chemical response of two varieties of grapes (Victoria and Shine Muscat) to the infection by Botrytis cinerea a widely distributed plant pathogen that attacks many economically important plants. The author analyzed volatile and non-volatile components by a series of instrumental methods coupled to statistical analyses. The paper is in the overall well written, except for some minor misprints. The experimental part is well executed, and the discussion of the results are consistent with previous result already published in the area of plant-pathogen interaction and the role of secondary plant products as a defense mechanism. Some minor errors should be addressed. The major observation is the lack of a -Conclusions- section that permits to the reader to integrate the results and to have a perspective for future related research project.

1.- In line 73, the initial letter of -solarbio science & technology- must be in uppercase to be in concordance with other company names included in the Material and Methods section.

Response: Corrected.

2.- In line 145 the word -sheath- must be replaced for the adequate term.

Response: We use a standard descriptive language on this part, and feel the “sheath” is adequate term in the LC/MS method [1]. Meanwhile, we modified the paragraph as follow:

LC-MS chromatographic analyses were performed on a Thermo Vanquish UHPLC system coupled with an ACQUITY UPLC HSS T3 (2.1 mm× 150 mm, 1.7µm) column and Thermo QE HF-X ESI ion. For positive ion analysis, the eluent was a mixture of methanol and H2O at a flow rate of 0.3 mL/min. The mobile phase consisted of 0.1% formic acid/H2O (solvent A), 0.1% formic acid /Methanol (solvent B), 0.1 acetic acid/H2O (solvent C), 0.1% acetic acid/Methanol (solvent D). Gradient steps for positive ion analysis were applied as follows (using solvents A and B): 0 to 0.5 min, B: 2%; 0.5 to 6 min, B: 2% to 50%; 6 to 10min, B: 50% to 98%; 10 to 14 min, B: 98%; 14 to 16 min, B: 98% to 2%; 16 to 21 min, B: 2%. For negative ion analysis, solvent C was used to replace A, and D to replace B. The MS system was operated using both positive and negative ESI in multiple reaction monitoring (MRM) mode to identify the analytes of interest. The operational parameters for MS analysis were as follows: ESI ion source; sheath gas flow at 50 L/min and auxiliary gas at 13 L/min, spray voltage: at 2.5KV (+)/2.5KV (-), capillary temperature maintained at 325 ℃; auxiliary gas temperature at 300 ℃; secondary collision energy (NCE) at 30V; isolation window at 1.5 m/z, top N = 10, scan range from 70 to 1050 m /z. The detected metabolites were qualified by using CD software, and analyzed at Thermo Fisher database, NIST database, and a self-built database.

3.- In line 99, Being…must be changed to Beijing?

Response: Corrected.

4.- A -Conclusions- section must be incorporated at the end of the paper. This section is of major relevance in a scientific paper.

Response: The conclusions section was added at lines 377-386:

“In this study, we have found that B. cinerea infection triggered grapes to increase the production of some secondary metabolites which have antifungal bioactivities and therefore enhanced plant resistance to diseases such as B. cinerea. We identified indicator compounds in grapes during the infection and discovered key secondary metabolites that are associated with the defensive activities of grapes in inhibiting B. cinerea. The downside of these PSMs is that the pathogen may be adaptive to and detoxify PSMs by enhancing the ABC efflux system, which impede the antimicrobial activity of PSMs. These findings enhance our understanding on the plant resistance to in host plants provided a new perspective for the development of MDR of B. cinerea. The PSM-adaptation in the pathogen should be considered when a fungicide is to be developed using PSMs and their derivatives.

[1] Lorkiewicz, P.K., Gibb, A.A., Rood, B.R., He, L., Zheng, Y., Clem, B.F., Zhang, X., Hill, B.G., 2019. Integration of flux measurements and pharmacological controls to optimize stable isotope-resolved metabolomics workflows and interpretation. Sci. Rep. 1–17. https://doi.org/10.1038/s41598-019-50183-3

Reviewer 3 Report

Wu et al described in this manuscript that the induction of secondary metabolites upon Botrytis cinerea infection. It is not surprised that the profiles changes overtime of pathogen infection and in different variety-strain combinations. The study is incremental to what have been already know about secondary metabolites induced by pathogen infection, however, it did not provide new insigts with grape- Botrytis cinerea interaction. As the title states ‘Dynamic changes of plant secondary metabolites induced by Botrytis cinerea infection contribute to plant defense to diseases and pathogen resistance to multiple drugs’. In fact, the title is misleading because there is no direct evidence in this study to show that changes in plant secondary metabolites contribute to plant defense and pathogen resistance to multiple drugs. The followings are other comments.

What does FC in tables stand for? What is the base of Log FC? 2?

Line 53. ‘Unfortunately, PSMs have controversial effects on invading pathogens?’ Is it ‘conflicting effects’? Please rephrase. This paragraph cited studies of insects, do insect detoxification mechanisms for adaptation reflect what happen to pathogens?

Line 54-55. Inaccurate claim…’some PSMs also induce pathogen and insect resistance to antibiotics and pesticides [12-14]’. References 12-14 all mentioned insects but pathogens?

Line 66-67. We have recently confirmed that PSMs can induce preadaptation-based multi-drug resistance (MDR) in the pathogen (Unpublished).

Line 90-91. Whether is Botrytis cinerea multidrug-resistant strain 242 while the strain B05.10 is drug-sensitive?

Line 135-236. Is eugenol (1.21-2.11 folds) considered as significant changes, give the variation nature of the assays?

Line 199 and 209. How are the arbitrary absolute peaks (height higher than 200,000) for GC-MS or 2,000 for LC-MS determined? Would it be possible some (maybe many) under those thresholds are also important metabolites but missed for identification as a results of Botrytis cinereal-grape interaction?

Table S1. Why a subset of the metabolites is regulated differently when infected with Botrytis cinerea B05.10 (B5), or with B. cinerea 242 (B2), given ‘Victoria’ is susceptible to both? The following are among those: Palatinose, 2,6-Dihydroxybenzoic acid, Phosphoric acid, D-(+)-Talose, 4-Ketoglucose, Cyclohexasiloxane, Copper phthalocyanine. In contrast, Table S2 showed majority metabolites except 2 phenols identified in the resistant ‘Shine Muscat’ are regulated in opposite between strains B05.10 and 242. The authors should address if multidrug-resistance of the strain 242 play different roles in compatible and incompatible interactions?

susceptible variety ‘Victoria’

resistant variety ‘Shine Muscat

Line 198-200. Only 206 out of 287 to 312 compounds deconvolved were identified, why are the remaining metabolites not identified?

Line 222. What is a tree treatment?

Line 234-235. resveratrol (2.68-13.72 folds)?

What about the metabolites that overlaps using both liquid chromatography/quadrupole time-of-flight mass (LC/QTOF) (line 230-236) vs Gas chromatography / quadrupole time-of-flight mass (GC/ QTOF)? It appears to be the case that the metabolite species are quite different depending on different methods used to detect. Which one is more reliable? What is the advantage or disadvantage for either LC/QTOF or GC/ QTOF? One is favored over the other? It is kind of more informative with the data from Table S1 and S2, while most of metabolites shown in table 2 are of little changes.

Line 244-245. ‘…but non-inoculated leaves were symptomless’, isn’t it obvious?

Phytoalexins detected?

Line 247. How were ‘133 metabolites selected and normalized? Why 133?

Line 299-300. Exaggerated statement. In fact, teugenol, 1.21-2.11; Salicylic acid, 0.9-1.98; Reserpine 0.16-1.19 shown in Table 2 suggested most of them just moderately regulated considering the variation of metabolites analysis.

Line 303-304. Ref. 31 is a review. Ref. 32 was a study analyzing trans-piceid, cis-piceid, trans-resveratrol, and cis-resveratrol in grape juice and proved these compounds are present without any induction. In any way, how were your results supported with their discoveries?

Line 341 Plant secondary metabolites (PSMs) appeared many times. Change to PSMs

Fig. 3. Is a different method used than the spraying used for non-volatiles? If Grape ‘Victoria’ (sensitive to B. cinerea) and ‘Shine Muscat’ (resistant to B. cinerea), why both show similar severe symptoms?

Fig. 4. It looks like no difference at all for ‘Victoria' between the inoculated with B. cinereal and non-inoculated.

Table 5. Did the experiments have biological replica and statistics analysis?

Ref 31. https://doi.org/10.1007/s00018-016-2415-7 not https://doi.org/1577-1603.10.1007/s00018-016-2415-7. Such mistakes occurred somewhere else in the text, please to make sure the cited papers are accurate.

The authors need to improve how they describe their experiments in an accurate and rational manner. Also needed to improve the interpretation of  the results. 

Author Response

Wu et al described in this manuscript that the induction of secondary metabolites upon Botrytis cinerea infection. It is not surprised that the profiles changes overtime of pathogen infection and in different variety-strain combinations. The study is incremental to what have been already know about secondary metabolites induced by pathogen infection, however, it did not provide new insights with grape- Botrytis cinerea interaction. As the title states ‘Dynamic changes of plant secondary metabolites induced by Botrytis cinerea infection contribute to plant defense to diseases and pathogen resistance to multiple drugs’. In fact, the title is misleading because there is no direct evidence in this study to show that changes in plant secondary metabolites contribute to plant defense and pathogen resistance to multiple drugs. The followings are other comments.

Response: Our study focused on analyzing the impact of pathogen infection on the plant metabolome, with a particular interest in identifying antibacterial compounds that could induce the up-regulation of ABC transporter genes from the up-regulated PSMs. Our objective was to shed light on the role of plant toxins in the co-evolution of fungicide resistance in pathogens. Our findings provide an important reference for the analysis of the causes of fungicide multidrug resistance in the field. Therefore, the title was changed to “Dynamic Changes in Plant Secondary Metabolites Induced by Botrytis cinerea Infection offer a Potential for Plant Defense to Diseases and Pathogen Resistance to Multiple Drugs”.

What does FC in tables stand for? What is the base of Log FC? 2?

Response: FC stands for “Fold change” and “FC, Fold change” was added at line 250. The base of Log FC was 2 and the mistake was corrected at line 270.

Line 53. ‘Unfortunately, PSMs have controversial effects on invading pathogens?’ Is it ‘conflicting effects’? Please rephrase. This paragraph cited studies of insects, do insect detoxification mechanisms for adaptation reflect what happen to pathogens?

Response: Despite extensive research on drug resistance in pathogens, the mechanism of pre-adaptation remains unclear. Conversely, research on drug resistance in insects has been well documented. It's important to note that the detoxification mechanism of insects may not necessarily reflect the resistance mechanism of pathogens. Although the preadaptation of pests primarily relies on detoxification metabolism, this does not fully explain the contribution of detoxification metabolism in the preadaptation of pathogens. In this paper, our data suggest that the preadaptation of pathogens is associated with efflux, and the role of detoxification metabolism in this process remains unclear.

Line 54-55. Inaccurate claim…’some PSMs also induce pathogen and insect resistance to antibiotics and pesticides [12-14]’. References 12-14 all mentioned insects but pathogens?

Response: The two sentences were combined and changed to “Unfortunately, PSMs have bidirectional effects on invading pathogens: some PSMs have antimicrobial activities, while some PSMs induce pathogen and insect resistance to antibiotics and pesticides.”

The insect detoxification mechanisms for adaptation reflect what has happened to plant pathogens.

The reference of pathogens were added as:

“14. Meirelles, L.A.; Perry, E.K.; Bergkessel, M.; Newman, D.K. Bacterial defenses against a natural antibiotic promote collateral resilience to clinical antibiotics. PLoS Biol 2021, 19(3), e3001093. https://doi.org/10.1371/journal.pbio.3001093

Line 66-67. We have recently confirmed that PSMs can induce preadaptation-based multi-drug resistance (MDR) in the pathogen (Unpublished).

Response: The sentence was modified as:

“We have recently confirmed that PSMs can induce preadaptation-based multi-drug resistance (MDR) in B. cinerea (Unpublished).”

Line 90-91. Whether is Botrytis cinerea multidrug-resistant strain 242 while the strain B05.10 is drug-sensitive?

Response: Yes, the sentence was modified as:

Botrytis cinerea standard-strain B05.10 was obtained from Germany.”

Line 135-236. Is eugenol (1.21-2.11 folds) considered as significant changes, give the variation nature of the assays?

Response: Eugenol was screened by one way anova and can be considered as significantly different.

Line 199 and 209. How are the arbitrary absolute peaks (height higher than 200,000) for GC-MS or 2,000 for LC-MS determined? Would it be possible some (maybe many) under those thresholds are also important metabolites but missed for identification as a results of Botrytis cinerea-grape interaction?

Response: Peak area is a widely used measurement for metabolome analysis. Instead of using a peak area as screening conditions, we set up a threshold, which is more suitable in the assay, and avoids missing any potential compounds. The listed peak area thresholds represent the lowest detection limits achievable under the instrument’s the conditions, which peak heights equivalent to approximately three times the noise level. It's worth noting that the assay may have unavoidable limitations, and undetected metabolites may not be excluded.

Table S1. Why a subset of the metabolites is regulated differently when infected with Botrytis cinerea B05.10 (B5), or with B. cinerea 242 (B2), given ‘Victoria’ is susceptible to both? The following are among those: Palatinose, 2,6-Dihydroxybenzoic acid, Phosphoric acid, D-(+)-Talose, 4-Ketoglucose, Cyclohexasiloxane, Copper phthalocyanine. In contrast, Table S2 showed majority metabolites except 2 phenols identified in the resistant ‘Shine Muscat’ are regulated in opposite between strains B05.10 and 242. The authors should address if multidrug-resistance of the strain 242 play different roles in compatible and incompatible interactions?

Response: The resistant strain (242) may have a resistance cost compared with the sensitive strain (B05.10), so it may be different from the sensitive strain in terms of infection power, but this result is not the main research object of this experiment, and the compound difference between the resistant strain and the sensitive strain before and after infection was also added in the discussion at line 353-362 as:

“Interestingly, some metabolites were regulated differently when the grape was infected by wild-type B. cinerea or by MDR-containing strain. While B. cinerea 242 can develop MDR, it may have cost of fitness and reduced virulence toward host plant, which was confirmed not only in this study, but also in several other studies on B. cinerea. Therefore, the acquisition of high-level resistance is associated with a decrease in fitness [53]. Meanwhile, grape plants with different resistance have different metabolomic activities as shown in the susceptible variety ‘Victoria’ versus the resistant variety ‘Shine Muscat.’ This indicates that plant defense against pathogen infection through metabolomic regulation is dependent on its resistance level.”

Line 198-200. Only 206 out of 287 to 312 compounds deconvolved were identified, why are the remaining metabolites not identified?

Response: Only a few compounds that have been accurately identified with secondary mass spectrometry of standard or reference. Due to the limitation of libraries, the rest compounds were not identified.

Line 222. What is a tree treatment?

Response: The sentence was modified as:

“There were 17 differential metabolites in ‘Shine Muscat’ under the treatments.”

Line 234-235. resveratrol (2.68-13.72 folds)?

Response: Corrected.

What about the metabolites that overlaps using both liquid chromatography/quadrupole time-of-flight mass (LC/QTOF) (line 230-236) vs Gas chromatography / quadrupole time-of-flight mass (GC/ QTOF)? It appears to be the case that the metabolite species are quite different depending on different methods used to detect. Which one is more reliable? What is the advantage or disadvantage for either LC/QTOF or GC/ QTOF? One is favored over the other? It is kind of more informative with the data from Table S1 and S2, while most of metabolites shown in table 2 are of little changes.

Response: GC-MS has advantages in the analysis of volatile substances (or derivatized) that are supported by a database. The two kinds of instruments complement each other. Most of the compounds in Table 2 have been reported to inhibit plant pathogens. We tend to explore compounds that have potential effects on inhibiting pathogens by using different measurements.

Line 244-245. ‘…but non-inoculated leaves were symptomless’, isn’t it obvious?

Response: The sentence has been modified as:

“Grape leaves inoculated with B. cinerea showed obvious disease symptoms compared to non-inoculated leaves.”

Phytoalexins detected?

Response: Yes, phytoalexins such as resveratrol were detected.

Line 247. How were ‘133 metabolites selected and normalized? Why 133?

Response: The 133 metabolites were selected after excluding outliers and qualitative analysis vis NIST14 and Fiehn mass spectrometry databases. The sentence was modified as:

“Among them, 133 metabolites were selected and normalized by excluding outliers and qualitative analysis vis NIST14 and Fiehn mass spectrometry databases in each group of sample.”

Line 299-300. Exaggerated statement. In fact, eugenol, 1.21-2.11; Salicylic acid, 0.9-1.98; Reserpine 0.16-1.19 shown in Table 2 suggested most of them just moderately regulated considering the variation of metabolites analysis.

Response: The sentence was modified as:

“We have found a trend that B. cinerea infection induced the production of several PSMs to be up-regulation than non-inoculated grape leaves, although the increase was moderate.”

Line 303-304. Ref. 31 is a review. Ref. 32 was a study analyzing trans-piceid, cis-piceid, trans-resveratrol, and cis-resveratrol in grape juice and proved these compounds are present without any induction. In any way, how were your results supported with their discoveries?

Response: Stilbene compounds are naturally present in plants, especially grape tissues. The infection by pathogen will cause the content of these compounds to be up-regulated, which is consistent with the reports in the literature.

Line 341 Plant secondary metabolites (PSMs) appeared many times. Change to PSMs

Response: Corrected.

Fig. 3. Is a different method used than the spraying used for non-volatiles? If Grape ‘Victoria’ (sensitive to B. cinerea) and ‘Shine Muscat’ (resistant to B. cinerea), why both show similar severe symptoms?

Fig. 4. It looks like no difference at all for ‘Victoria' between the inoculated with B. cinereal and non-inoculated.

Response: The inoculation method in Fig 3 with Botrytis cinerea was modified, shown at lines 154-156:

“The petioles of grape leaves were inserted into the wet cotton and the 5 mm mycelial disks of B. cinerea were inoculated with.”

We used the same method for testing both volatile and no-volatile compounds.

The appearance of symptoms on ‘Shine Muscat’ later than ‘Victoria’ and was not significantly differenct between two varieties of grapes. At day 3, lesions were observed on ‘Victoria’, but not on ‘Shine Muscat’. At day 6, lesions were also observed on ‘Shine Muscat’ and larger than that on ‘Victoria.’ The time of disease development was longer after 9 days, but there was no significant difference in the area of lesion between the two groups.

The appearance of symptoms on ‘Shine Muscat’ was delayed compared to ‘Victoria’ and there was less difference between the two varieties of grapes in the end of the observation.

Table 5. Did the experiments have biological replica and statistics analysis?

Response: The experiment was conducted twice, and each experiment included three replicates for each treatment. In addition, Table 5 was changed to Fig 6 and the statistical analysis was visualized in the figure.

Ref 31. https://doi.org/10.1007/s00018-016-2415-7 not https://doi.org/1577-1603.10.1007/s00018-016-2415-7. Such mistakes occurred somewhere else in the text, please to make sure the cited papers are accurate.

Response: It has been corrected and all references were checked.

The authors need to improve how they describe their experiments in an accurate and rational manner. Also needed to improve the interpretation of the results.

Response: We went through the manuscript and made some changes.

Round 2

Reviewer 1 Report

It is acceptable in the present form

Author Response

Thanks for the reviewer. 

Reviewer 3 Report

The new title looked more convoluted. My suggestion to keep it simple and concise: Dynamic Changes in Plant Secondary Metabolites Induced by Botrytis cinerea Infection. The rest can be discussed in the paper.

The overall English is OK.

Author Response

We agree to change the title as the reviewer suggested: Dynamic Changes in Plant Secondary Metabolites Induced by Botrytis cinerea Infection.